# Clinical questions in primary care: Where to find the answers - a cross-sectional study

**Catarina Viegas Dias**[1,2]*, **Clara Jasmins**[2], **David Rodrigues**[1,2], **Bruno Heleno**[1,2]

**1** CHRC, NOVA Medical School|Faculdade de Ciências Médicas, NMS|FCM, Universidade Nova de Lisboa, Lisboa, Portugal, **2** NOVA Medical School|Faculdade de Ciências Médicas, NMS|FCM, Universidade Nova de Lisboa, Lisboa, Portugal

* catarina.viegasdias@nms.unl.pt

## Abstract

### Introduction

Clinicians raise at least one question for every two patients they see, but search for an answer to less than half of these questions, and rarely use evidence-based resources. One barrier to evidence-based practice is doubt that the search would yield an answer, and we found insufficient evidence to refute this concern. This study aims to identify what proportion of clinical questions in primary care can be answered with online evidence-based practice resources, and what proportion of these can be answered with pre-appraised evidence.

### Materials and methods

Cross-sectional study in two primary care practices. The inclusion criteria were family doctors, generalists and residents working in 2 selected practices. We collected a total of 238 questions from 19 family medicine specialists, 9 family medicine residents and 3 generalist doctors. Doctors were asked to record any clinical question that arose during 4 days of appointments. The primary outcome was the proportion of clinical questions answered with online evidence-based practice resources. The secondary outcome was the level of evidence needed to reach to find the answers (clinical summaries, systematic guidelines, systematic reviews or primary studies), according to Haynes' pre-appraised evidence pyramid model.

### Results

191 of the 206 valid clinical questions could be answered with online evidence-based practice resources (92.7% [95% CI 88.3%-95.9%]). Most of these questions (90.8% CI 95% 85.9%-94.4%) were successfully answered using clinical summaries (BMJ Best Practice, DynaMed or UpToDate), with a median search time of 4 minutes (range 1–16.5).

### Conclusions

Contrary to clinician's beliefs, the majority of clinical questions can be answered with online evidence-based practice resources, and most of them with pre-appraised evidence. This study could encourage family doctors to increase the use of clinical summaries.

**Data Availability Statement:** The dataset for this study is deposited in a repository (Zenodo), available through the following DOI: 10.5281/zenodo.7275128.

**Funding:** The present publication was funded by Fundação Ciência e Tecnologia, IP national support through CHRC (UIDP/04923/2020).

**Competing interests:** All the researchers are teachers in undergraduate and postgraduate courses on Evidence Based Medicine and Critical Appraisal of Medical Literature at NOVA Medical School. This does not alter our adherence to PLOS ONE policies on sharing data and materials.

Furthermore, these results highlight the importance of teaching how to search for and apply pre-appraised evidence.

## Introduction

Clinicians raise at least one question for every two patients they see, but search for an answer to less than half of these questions [1], and rarely use evidence-based resources [2–4]. Barriers to evidence-based practice include insufficient time, not knowing how to select an optimal search strategy, and doubt that the search would yield an answer [5, 6]. To the best of our knowledge, the evidence about this topic is too scarce and indirect to assess whether such belief is justified. One 2004 paper found that evidence-based practice resources completely answered 20.0% of the 40 questions asked, but it included only complex clinical questions [7]. In another study, residents were able to answer between 45% and 86% of a standard set of clinical questions using different evidence-based practice resources [8]. In both studies, the questions were not elicited by the clinicians, but rather chosen by the researchers, and these may differ from questions emerging in everyday practice.

The question still remains: can clinicians be reassured that their questions have readily available evidence-based answers?

Furthermore, clinicians seem to have a 90 second rule—they don't try to search for an answer if they judge it to take more than a minute and a half to find [9], and tend to prefer answers from pre-appraised evidence [10]. Hence, it would be relevant to identify not only the proportion of questions answered by evidence-based resources, but whether they were found in pre-appraised evidence or in more time-consuming resources.

Haynes et al. proposed a pyramid model for finding pre-appraised evidence (Fig 1), with five levels: primary studies, systematic reviews, guidelines, clinical summaries and systems [11]. Each of these levels should build systematically from the previous level and provide more summarized and more useful information for guiding clinical decision-making. When available, resources higher up the pyramid should be more efficient for daily clinical practice. Since systems, the most synthetized and "automatic" resource (the evidence synthesis would be integrated in the Electronic Health Records and linked automatically to each individual patient) are still proving challenging to implement, the most synthetized resources available for most clinicians nowadays are clinical summaries. Clinical summaries consist of frequently updated syntheses of evidence, accompanied by systematically derived recommendations such as British Medical Journal (BMJ) Best Practice, DynaMed Plus, Evidence-Based Medicine (EBM) Guidelines, Essential Evidence Plus and UpToDate. The use of the Haynes' evidence-based pyramid model could help increase search efficiency and be more realistic in daily clinical practice. However, to our knowledge, this method is far from being standard practice in searching for answers in Primary Care.

This cross-sectional study aims to identify what proportion of clinical questions in primary care can be answered with online evidence-based practice resources, and what level of the 5S pyramid is needed to reach to find an answer [11].

## Materials and methods

### Design

This was a cross-sectional study. We followed the STROBE statement for reporting observational studies.

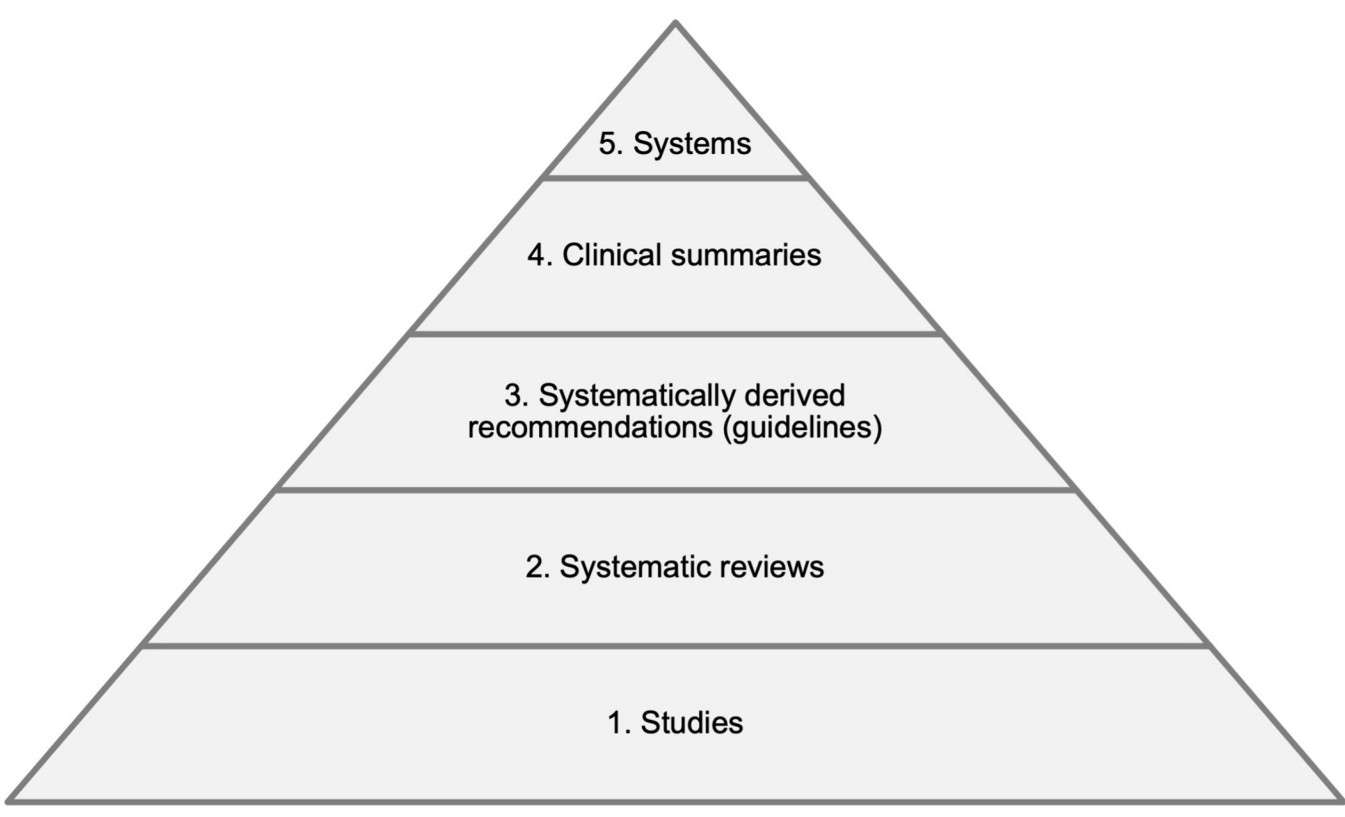

**Fig 1. Evidence-based healthcare pyramid for finding pre-appraised evidence.**

### Ethics

The protocol was approved by the local Ethics Committee (Lisbon and Tagus Valley Regional Health Administration), and participants signed a written informed consent.

### Setting

The study was conducted at 2 primary care practices in Lisbon, Portugal. Portuguese primary care practices are publicly funded and are staffed by teams of around 4–10 family doctors or general practitioners, 4–10 practice nurses, 2–6 clinical secretaries and a varying number of family medicine residents, foundation year doctors and nursing students. Both participating practices are academic practices, involved in undergraduate and vocational training of health professionals.

### Participants

The defined inclusion criteria were: family doctors and family medicine residents who signed the informed consent.

Regarding sample size, there were no previous estimates of the proportion of questions answerable by online evidence-based practice resources. Thus, a conservative estimate of 50% for the expected proportion was used, which resulted in a total sample of 385 questions to ensure that the 95% confidence interval estimate would be within the 5% of the true proportion.

Assuming that Portuguese family doctors have about 19 appointments per day [12] and at least one question for every two patients they see [1], 10 clinical questions per doctor per day

were expected. Since the two practices had 36 doctors, a participation rate of 60% (n = 21) would yield 210 questions per study day.

We were concerned that doctors could have less than the average number of appointments during the study period, or that they could not recall (or be willing to share) all their clinical questions. As such, we anticipated only half (5) clinical questions per doctor per day. Since the two pilot practices had 36 doctors, a participation rate of 60% (n = 21) would yield 105 questions per study day and 4 days would be needed to reach the target sample size.

Participants were recruited during January 2020. Data was collected through a questionnaire which included baseline data (age, licensing status and years of clinical experience), number of appointments per day, and any clinical questions that arose during one day of clinical practice. Demographic and professional data were also collected from non-participants, to allow assessing for participation bias while maintaining anonymity.

## Procedures

All doctors in the two practices were asked about their age, licensing status and years of clinical experience since graduation.

After recruitment, two researchers (CVD and CJ) provided participants with an anonymized data collection form in which they could record all clinical questions that were raised during consultations, instead of directly interviewing them after each appointment, in order to avoid social desirability bias. They were asked to record any clinical questions that arose during the appointment, omitting any patient identifiable information, as well as the number of appointments per day.

Two researchers then independently classified the type of clinical question as background questions (which ask for general knowledge about a condition, test or treatment) or foreground questions (which ask for specific knowledge to inform clinical decisions). Foreground questions were further classified as differential diagnosis, diagnostic accuracy, treatment, etiology / harm, prognosis, patient experiences, as defined by Guyatt et al. [13]. The two researchers then proceeded to search online evidence-based practice resources to answer the questions elicited by clinicians. The search was initiated in 3 clinical summaries databases, BMJ Best Practice, DynaMed and UpToDate. These 3 databases were chosen since the most updated review of web-based point-of-care summaries ranked these highest in terms of breadth of disease coverage, editorial quality and evidence-based methodology [14]. If no answer was found in these resources, the researchers proceeded to search for systematic guidelines using guideline databases (GuidelineCentral, GRADE guidelines, NICE guidelines and SIGN guidelines websites) and federated search databases which allow the selection of a clinical guidelines filter (ACCESSSS, TRIP database and Epistemonikos). If no adequate answer was found, federated search databases were used to search for synopses of systematic reviews and sequentially, synopses of primary studies. If no evidence was found in pre-appraised evidence, the researchers searched for systematic reviews using federated search databases, as well as MEDLINE and Cochrane Library, and sequentially, primary studies in federated search databases and MEDLINE. The time for each search was recorded by both researchers, and a time limit of 30 minutes was used, given the point-of-care context. The 2 researchers then independently recorded the level of the 5S evidence-based pyramid needed to reach to find a clinically useful answer. Adequacy of answers was defined by consensus of two researchers who are general practitioners, and disagreements were solved by a third senior researcher.

Since clinical adequacy of the answers is a subjective outcome, a post-hoc analysis was performed to test the robustness of the primary analysis. Two blinded medical assessors who were not part of the research team independently evaluated if the answers found by the researchers

were considered adequate to inform decision-making. We then performed a conservative analysis, considering any answer classified by a blinded outcome assessor as "not informative" as an answer not found in the available resources, and only the answers classified by both assessors as "informative" as answers found in the available resources. Another post-hoc analysis was performed to check for correlation between physician experience and proportion of background questions.

Finally, one researcher registered the strength of the recommendation behind each answer, rated by clinical summaries and guidelines, and presented it according to the Grading of Recommendations Assessment, Development and Evaluation (GRADE) approach as strong, weak/conditional, or not classified.

## Main outcome measures

The primary outcome was presented as a proportion, with 95% confidence intervals. We presented narratively how many questions were answerable at each level of the evidence pyramid. Search time was reported through descriptive statistics. Correlation between years of experience and formulation of foreground questions was calculated using Spearman's coefficient. Analyses were performed on Stata v 14.1.

## Patient and public involvement

No patients were involved in designing or conducting this study.

## Results

We collected 238 questions from 31 clinicians (participation rate of 86.1%, total of 36 clinicians), from which 16 questions were classified as non-clinical and 16 as incomplete/miswritten, leaving a total of 206 valid clinical questions. 132 questions were background questions and 74 were foreground questions. Among the latter, 15 were differential diagnosis questions, 4 were diagnostic accuracy questions, 7 were etiology/harm questions, 4 were prognosis questions, 44 were treatment/intervention questions and none was classified as patient experience.

Participant characteristics are reported in Table 1.

Regarding the primary outcome, 191 of the 206 clinical questions were answered with online evidence-based practice resources, resulting in a proportion of 92.7% (CI 95% 88.3%-95.9%). Only 15 clinical questions remained unanswered after going through the 5 levels of pre-appraised evidence-based practice resources and systematic reviews and primary studies databases (12 background questions and 3 foreground questions), within the 30-minute limit previously defined.

Most clinical questions (187 of 206, 90.8% CI 95% 85.9%-94.4%) were successfully answered using level 2 resources (clinical summaries, namely, BMJ Best Practice, DynaMed or UpToDate). Three questions were answered using guidelines (level 3). Within the 30-minute

**Table 1. Age and experience distribution.** Participants' age was reported in 10-year categories to maintain anonymity, as well as clinical experience (years practicing with autonomy); the number of participants in each age and experience category is registered under frequency (total sample = 31).

| Age category (years) | Frequency | Experience category (years) | Frequency |
|---|---|---|---|
| 20–29 | 2 | <5 | 8 |
| 30–39 | 12 | 5–9 | 7 |
| 40–49 | 3 | 10–19 | 4 |
| 50–59 | 6 | >20 | 12 |
| >60 | 8 | | |

time limit, no question was answered by systematic review synopses (level 4) or primary studies synopses (level 5), but one answer was found in a primary study (non-previously appraised).

The post-hoc analysis conducted by two medical assessors blinded to the study objectives yielded a more favorable result than the primary analysis: 198 of the 206 answers considered informative for decision-making (96.1%, CI 95% 92.5%-98.3%).

Regarding the strength of recommendations, among the 74 foreground questions, 11 were answered with strong recommendations, 18 were answered with conditional/weak recommendations, 21 were not classified by the resource and 24 were considered not applicable to this classification (due to multiple recommendations within the same answer).

We found that most of background questions (99/132) were made by clinicians with less than 10 years of experience, and we found a moderate correlation between years of experience and formulation of foreground questions (Spearman's coefficient 0.56, 95%CI 0.46 to 0.65).

Regarding the time needed to find an answer, an exploratory analysis found that the median value was 4 minutes (range 1–16.5) for the total of answered questions (n = 191), and 3.5 minutes (range 1–13.5) for questions answered by clinical summaries. The distribution of search time is presented in Fig 2.

## Discussion

Our study suggests most clinical questions raised by primary care clinicians can be answered with pre-appraised evidence. These results could encourage family doctors to use point of care clinical summaries to increase evidence-based decision-making in Primary Care.

The interpretation of these results is limited by a non-representative sample (only 2 practices, and both involved in undergraduate training). In addition, questions reported by participants were "translated" into a PICO (Patients, Intervention, Comparison, Outcome) by the researchers. To keep participant anonymity, researchers had no way of confirming with participants whether the translation was an accurate formulation of the original question. Although most answers were considered clinically useful by both researchers and blind medical assessors, only a minority of answers had a clear strength of recommendation stated in the evidence-based resource from which they were retrieved.

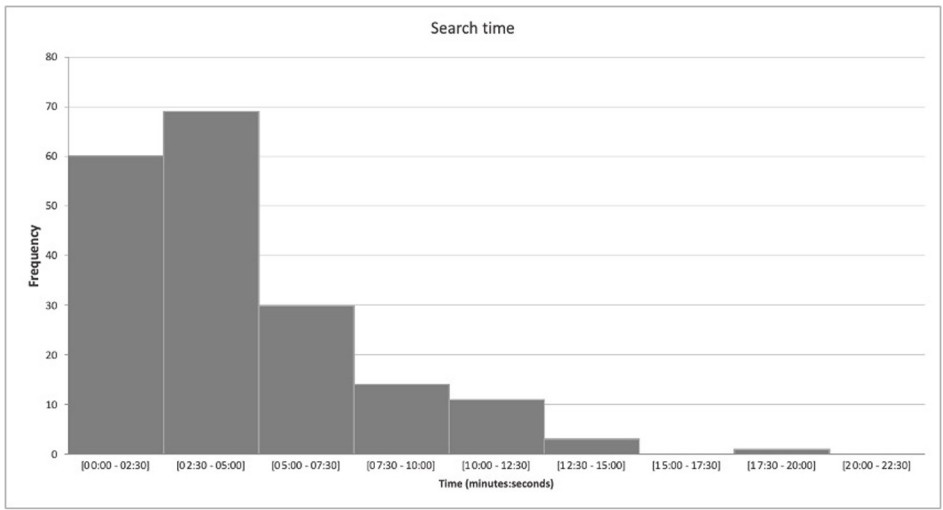

**Fig 2. Search time distribution.** Mean search time (2 researchers), in minutes.

Despite limitations, these results are exciting news in a point-of-care context. Two of the most cited barriers to pursuing clinical questions are "doubt that a useful answer exists" and "lack of time" [3]. This study contributes with quantitative evidence that the first belief is false. As clinicians prefer pre-appraised evidence [5] and spend a mean of less than 2 to 3 minutes seeking an answer to a specific question [6], our results may also help address the second belief.

Although the majority (67,9%) of the questions was answered in less than 5 minutes, there was still a considerable amount of questions that took more than 5 minutes to answer, which could decrease the feasibility of this method. The mean search time of researcher 2 was significantly shorter than researcher 1, who was less experienced in evidence-based practice, and the mean search time decreased in the second half of questions for both researchers. This exploratory analysis on the researchers' search time suggested that practicing searching for evidence-based answers decreases the time needed for the searches. This is consistent with a recent systematic review that suggests search time can be optimized with practice [15]. However, since this was not a main outcome of the study, these results should be interpreted with caution.

Future research could focus on determining time needed to find an answer as a pre-specified analysis. Furthermore, it would be interesting to assess if choice architecture (e.g., nudges such as using automatic sign in as a default, "infobuttons" that automate the search against a set of pre-defined resources) could increase the use of online evidence-based resources in Primary Care.

Regarding medical education, this study highlights the importance of teaching how to search for and apply pre-appraised evidence. Evidence-based medicine courses curricula focus greatly in critically appraising several types of primary studies [16], and this might explain why some family physicians perceive evidence-based medicine to be of limited value in primary care [17]. Lack of time and expertise to analyze primary studies and decide whether to apply them to clinical practice is frequently cited in qualitative studies and is deemed impractical by family physicians [16]. The use of the Haynes' evidence-based pyramid model could help increase search efficiency and be more realistic in daily clinical practice. However, to our knowledge, this method is far from being standard practice in searching for answers in Primary Care.

## Conclusions

Contrary to clinician's beliefs, the majority of clinical questions can be answered with online evidence-based practice resources, and most of them with pre-appraised evidence. This study could encourage family doctors to increase the use of clinical summaries. Furthermore, these results highlight the importance of teaching how to search for and apply pre-appraised evidence.

## Acknowledgments

Pascale Charondière, MD; Paula Broeiro, PhD; Rita Lopes da Silva, MD; Mariana Martinho, MD.

## Author Contributions

**Conceptualization:** Catarina Viegas Dias, Clara Jasmins, David Rodrigues, Bruno Heleno.

**Data curation:** Catarina Viegas Dias, Clara Jasmins.

**Formal analysis:** Catarina Viegas Dias, Clara Jasmins, Bruno Heleno.

**Methodology:** Catarina Viegas Dias, Clara Jasmins, David Rodrigues, Bruno Heleno.

**Project administration:** Catarina Viegas Dias.

**Supervision:** Catarina Viegas Dias, David Rodrigues, Bruno Heleno.

**Validation:** Clara Jasmins.

**Writing – original draft:** Catarina Viegas Dias, Bruno Heleno.

**Writing – review & editing:** Catarina Viegas Dias, Clara Jasmins, David Rodrigues, Bruno Heleno.

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
