## [Decision Letter · Decision Letter 0]

13 Jul 2022

PONE-D-22-05275Clinical questions in primary care: where to find the answers - a cross-sectional studyPLOS ONE

Dear Dr. Viegas Dias,

Thank you for submitting your manuscript to PLOS ONE; I sincerely apologise for the unusually delayed review timeframe. Your manuscript has now been assessed by one reviewer, whose comments are appended below. After careful consideration, we feel that it has merit but does not fully meet PLOS ONE’s publication criteria as it currently stands. Although the reviewer commented that the manuscript was "well-written and the study adds to the knowledge about the potential utility and feasibility of using evidence-based resources in patient care", they raised several points regarding the description of the methodology and discussion of limitations. Therefore, we invite you to submit a revised version of the manuscript that addresses the points raised during the review process. Please note that we have only been able to secure a single reviewer to assess your manuscript. We are issuing a decision on your manuscript at this point to prevent further delays in the evaluation of your manuscript. Please be aware that the editor who handles your revised manuscript might find it necessary to invite additional reviewers to assess this work once the revised manuscript is submitted. However, we will aim to proceed on the basis of this single review if possible.

We look forward to receiving your revised manuscript.

Kind regards,

Emily Chenette

Editor in Chief

PLOS ONE

All researchers are teachers in undergraduate and postgraduate courses on Evidence Based Medicine and Critical Appraisal of Medical Literature at NOVA Medical School.”

Reviewers' comments:

Reviewer's Responses to Questions

**Comments to the Author**

1. Is the manuscript technically sound, and do the data support the conclusions?

Reviewer #1: Yes

2. Has the statistical analysis been performed appropriately and rigorously? 

Reviewer #1: Yes

3. Have the authors made all data underlying the findings in their manuscript fully available?

Reviewer #1: No

4. Is the manuscript presented in an intelligible fashion and written in standard English?

Reviewer #1: Yes

5. Review Comments to the Author

Reviewer #1: The authors collected 238 clinical questions that primary care physicians raised in ambulatory care and assessed whether these questions could be answered by evidence-based resources. They found that over 92% of the questions could be answered, primarily using clinical summaries, and in a relatively short time. The manuscript is well-written and the study adds to the knowledge about the potential utility and feasibility of using evidence-based resources in patient care. Suggestions to improve the manuscript are provided below.

1) Consider adding the search time measure to the abstract, since search time is critical to determine feasibility of pursuing those questions at the point-of-care.

2) Readers may not know the different between “background” and “foreground” questions, so it would be important to define in the methods.

3) In the results, the following statement is ambiguous: “no question was answered by systematic review synopses (level 4) and no question was answered by primary studies synopses (level 5)”. Were systematic reviews/primary studies actually searched for every single question? I assume that, based on the study methods, they were not. But it would be important to make it more clear.

4) As proposed above, the search time outcome is critical. Rather than just median and range, consider adding the entire distribution, perhaps including a histogram or a table. This could also have implications for the discussion. For example, even if most questions could be answered in less than 90 seconds, a relatively small percentage of questions taking longer than 5 minutes could be sufficient to discourage physicians who may be very concerned about pursuing an answer without knowing upfront how long it could take.

5) In the discussion, 4th paragraph, another potential nudge is “infobuttons” that not only bypass authentication but also automate the search against a set of pre-defined resources.

6) Was there any effort to validate that the answers found were sufficient to inform decision-making and supported by high-quality evidence? It appears that this determination (found vs. not found) was based on the subjective assessment of the co-authors who conducted the searches and were not blinded to the study objectives. This weakness is arguably more important than any of the limitations discussed in the manuscript.

6. PLOS authors have the option to publish the peer review history of their article (what does this mean?). If published, this will include your full peer review and any attached files.

Reviewer #1: No

---

## [Author Response · Author response to Decision Letter 0]

25 Aug 2022

Response to Editor:

We hereby submit the revised manuscript, cover letter and response to reviewers as requested, addressing the requirements cited in the email received by the corresponding author:

1. We followed PLOS ONE's style requirements;

2. We confirm there are no alterations to our adherence to all PLOS ONE policies on sharing data and materials, and included our updated Competing Interests statement in the cover letter;

3. We confirm that we will provide repository information for our data at acceptance. Our data will be upload to a public repository (Zenodo), as soon as we have access to the publication date, which is needed for uploading datasets in this platform;

4. We placed the ethics statement in the Methods section of our manuscript. 

Response to Reviewer:

We hereby respond to each point raised by the reviewer:

1. Is the manuscript technically sound, and do the data support the conclusions?

Reviewer #1: Yes

2. Has the statistical analysis been performed appropriately and rigorously?

Reviewer #1: Yes

3. Have the authors made all data underlying the findings in their manuscript fully available?

Reviewer #1: No

Thank you for your comment. Our data will be upload to a public repository (Zenodo), as soon as we have access to the publication date, which is needed for uploading datasets in this platform. 

4. Is the manuscript presented in an intelligible fashion and written in standard English?

Reviewer #1: Yes

5. Review Comments to the Author

Reviewer #1: The authors collected 238 clinical questions that primary care physicians raised in ambulatory care and assessed whether these questions could be answered by evidence-based resources. They found that over 92% of the questions could be answered, primarily using clinical summaries, and in a relatively short time. The manuscript is well-written and the study adds to the knowledge about the potential utility and feasibility of using evidence-based resources in patient care. Suggestions to improve the manuscript are provided below.

1) Consider adding the search time measure to the abstract, since search time is critical to determine feasibility of pursuing those questions at the point-of-care.

Thank you for your suggestion. We added the search time measure to the abstract.

2) Readers may not know the difference between “background” and “foreground” questions, so it would be important to define in the methods.

Thank you for your suggestion. We added the definition of “background” and “foreground” questions in the methods section.

3) In the results, the following statement is ambiguous: “no question was answered by systematic review synopses (level 4) and no question was answered by primary studies synopses (level 5)”. Were systematic reviews/primary studies actually searched for every single question? I assume that, based on the study methods, they were not. But it would be important to make it more clear.

Thank you for your comment. We clarified the statement to “most clinical questions (187 of 206, 90.8% CI 95% 85.9%-94.4%) were successfully answered using level 2 resources (clinical summaries, namely, BMJ Best Practice, DynaMed or UpToDate). Three questions were answered using guidelines (level 3). Within the 30-minute time limit, no question was answered by systematic review synopses (level 4) or primary studies synopses (level 5), but one answer was found in a primary study (non-previously appraised).”

4) As proposed above, the search time outcome is critical. Rather than just median and range, consider adding the entire distribution, perhaps including a histogram or a table. This could also have implications for the discussion. For example, even if most questions could be answered in less than 90 seconds, a relatively small percentage of questions taking longer than 5 minutes could be sufficient to discourage physicians who may be very concerned about pursuing an answer without knowing upfront how long it could take.

Thank you for your suggestion. We added a histogram representing the entire search time distribution in the results section.

Regarding the implications of these findings, we added the following paragraph to the discussion section:

“Although the majority (67,9%) of the questions was answered in less than 5 minutes, there was still a considerable amount of questions that took more than 5 minutes to answer, which could decrease the feasibility of this method. The mean search time of researcher 2 was significantly shorter than researcher 1, who was less experienced in evidence-based practice, and the mean search time decreased in the second half of questions for both researchers. This exploratory analysis on the researchers’ search time suggested that practicing searching for evidence-based answers decreases the time needed for the searches. This is consistent with a recent systematic review that suggests search time can be optimized with practice. However, since this was not a main outcome of the study, these results should be interpreted with caution.”

5) In the discussion, 4th paragraph, another potential nudge is “infobuttons” that not only bypass authentication but also automate the search against a set of pre-defined resources.

Thank you for your suggestion. We added this information in the discussion.

6) Was there any effort to validate that the answers found were sufficient to inform decision-making and supported by high-quality evidence? It appears that this determination (found vs. not found) was based on the subjective assessment of the co-authors who conducted the searches and were not blinded to the study objectives. This weakness is arguably more important than any of the limitations discussed in the manuscript.

Thank you for your comment, and for the identification of a relevant limitation that prompted us to conduct a sensitivity analysis. We asked two blinded medical assessors who were not part of the research team to evaluate if the answers found by the researchers were considered adequate to inform decision-making. We then performed a conservative analysis, considering any answer classified by one of the blinded outcome assessors as “not informative” as an answer not found in the available resources, and only the answers classified by both assessors as “informative” as answers found in the available resources. Contrary to our expectation, this sensitivity analysis yielded a more favorable result than the primary analysis (sensitivity analysis: 198 of the 206 answers considered informative for decision-making by both blinded medical assessors [96.1%, CI 95% 92.5%-98.3%]; primary analysis: 191 of the 206 answers considered informative for decision-making by the researchers [92.7%, CI 95% 88.3%-95.9%]). 

Regarding the quality of evidence, we registered the strength of the recommendation behind each answer, rated by clinical summaries and guidelines, and presented it according to the Grading of Recommendations Assessment, Development and Evaluation (GRADE) approach as strong, weak/conditional, or not classified. Considering the 74 foreground questions, 11 were answered with strong recommendations, 18 were answered with conditional/weak recommendations, 21 were not classified by the resource and 24 were considered not applicable to this classification (due to multiple recommendations within the same answer).

We added these analyses to the manuscript.

---

## [Decision Letter · Decision Letter 1]

26 Sep 2022

PONE-D-22-05275R1Clinical questions in primary care: where to find the answers - a cross-sectional studyPLOS ONE

Dear Dr. Viegas Dias,

Thank you for submitting your manuscript to PLOS ONE. After careful consideration, we feel that it has merit but does not fully meet PLOS ONE’s publication criteria as it currently stands. Therefore, we invite you to submit a revised version of the manuscript that addresses the points raised during the review process.

We look forward to receiving your revised manuscript.

Kind regards,

Guilherme Del Fiol

Guest Editor

PLOS ONE

Journal Requirements:

Additional Editor Comments (if provided):

Thank you for addressing the reviewer's comments, especially the extra effort collecting additional date. The revised manuscript has substantially improved. There is just one final minor suggestion regarding the legend of Table 1.

Reviewers' comments:

Reviewer's Responses to Questions

**Comments to the Author**

1. If the authors have adequately addressed your comments raised in a previous round of review and you feel that this manuscript is now acceptable for publication, you may indicate that here to bypass the “Comments to the Author” section, enter your conflict of interest statement in the “Confidential to Editor” section, and submit your "Accept" recommendation.

Reviewer #2: (No Response)

2. Is the manuscript technically sound, and do the data support the conclusions?

Reviewer #2: Partly

3. Has the statistical analysis been performed appropriately and rigorously? 

Reviewer #2: Yes

4. Have the authors made all data underlying the findings in their manuscript fully available?

Reviewer #2: Yes

5. Is the manuscript presented in an intelligible fashion and written in standard English?

Reviewer #2: Yes

6. Review Comments to the Author

Reviewer #2: Table 1 needs a more descriptive legend. Please describe each column, especially the 'Experience' column.

7. PLOS authors have the option to publish the peer review history of their article (what does this mean?). If published, this will include your full peer review and any attached files.

Reviewer #2: No

---

## [Author Response · Author response to Decision Letter 1]

25 Oct 2022

We hereby respond to the point raised by reviewer #2 and add a correction to the reference list.

Review Comments to the Author

Reviewer #2: Table 1 needs a more descriptive legend. Please describe each column, especially the 'Experience' column.

Thank you for your comment. We clarified the legend with a description of each column. 

Table 1. Age and experience distribution. Participants’ age was reported in 10-year categories to maintain anonymity, as well as clinical experience (years practicing with autonomy); the number of participants in each age and experience category is registered under frequency (total sample = 31).

Correction to reference list:

We noted there was a duplication of a reference and a formatting error in references 16 and 18. We unified the two references and clarified the cited article:

- Albarqouni L, Hoffmann T, Straus S, Olsen NR, Young T, Ilic D, et al. Core Competencies in Evidence-Based Practice for Health Professionals: Consensus Statement Based on a Systematic Review and Delphi Survey. JAMA Network Open. 2018 Jun 22;1(2):e180281.

---

## [Editor Report · Decision Letter 2]

28 Oct 2022

Clinical questions in primary care: where to find the answers - a cross-sectional study

PONE-D-22-05275R2

Dear Dr. Viegas Dias,

We’re pleased to inform you that your manuscript has been judged scientifically suitable for publication and will be formally accepted for publication once it meets all outstanding technical requirements.

Kind regards,

Guilherme Del Fiol

Guest Editor

PLOS ONE
---

## [Editor Report · Acceptance letter]

4 Nov 2022

PONE-D-22-05275R2 

Clinical questions in primary care: where to find the answers - a cross-sectional study 

Dear Dr. Viegas Dias:

I'm pleased to inform you that your manuscript has been deemed suitable for publication in PLOS ONE. Congratulations! Your manuscript is now with our production department. 

Kind regards, 

on behalf of

Dr. Guilherme Del Fiol 

Guest Editor

PLOS ONE